# Incorporation of Plant Extracted Hydroxyapatite and Chitosan Nanoparticles on the Surface of Orthodontic Micro-Implants: An In-Vitro Antibacterial Study

**DOI:** 10.3390/microorganisms10030581

**Published:** 2022-03-07

**Authors:** Anwar S. Alhazmi, Sohier M. Syame, Wael S. Mohamed, Ashraf S. Hakim

**Affiliations:** 1Department of Preventive Dentistry, College of Dentistry, Jazan University, Jazan 82912, Saudi Arabia; 2Department of Microbiology and Immunology, National Research Centre, Giza 12622, Egypt; sohiersyame@yahoo.com (S.M.S.); migris410@yahoo.com (A.S.H.); 3Department of Polymer, National Research Centre, Giza 12622, Egypt; wsabry1976@yahoo.com

**Keywords:** nano-HAP, oral pathogens, chitosan, *Salvadora persica*, titanium micro-implants

## Abstract

In our study, the structural and morphological applications of hydroxyapatite and chitosan nanoparticles and coated micro-implants were assessed for their ability to combat oral pathogenic bacteria. The hydroxyapatite, as well as chitosan nanoparticles, were synthesized from the *Salvadora persica* plant. The crystal morphology, phase composition, particle size, and surface functional groups of the nano-samples were analyzed via classical examinations and energy dispersive X-ray analysis. The prepared nanoparticles have been examined for antibacterial activity against four common oral bacterial strains. The antimicrobial effect was also assessed by the Live/Dead BacLight technique in combination with confocal scanning laser microscopy. Titanium micro-implants were coated with regular hydroxyapatite (HAP) and chitosan nanoparticles, and the surface was characterized by scanning electron microscopy. The analysis asserted elemental composition of the prepared nanoparticles and their textural features, metal crystallization, and functional bonds. The antibacterial activity of the nanoparticles was evaluated against oral pathogenic microorganisms by the disc diffusion method, minimum bacterial concentration (MBC), and minimum inhibitory concentration (MIC). Chitosan nanoparticles showed (MICs) of 8 μg mL^−1^ for (*Streptococcus salivarius*, *Streptococcus mutans* and *Enterococcus faecalis*), and 16 μg mL^−1^ for *Streptococcus sanguinis*. HAP nanoparticles showed (MICs) of 16 μg/mL for *E. faecalis*, and *S. sanguis*, 8 μg/mL for *S. salivarius* and finally 4 μg/mL for *S. mutans*. HAP nanoparticles showed enhanced antibacterial activity and more obvious damage in the bacterial cell membrane than that of synthesized chitosan nanoparticles. The prepared nanoparticles could successfully coat titanium microplates to enhance their efficiency.

## 1. Introduction

Micro-implants are often used in orthodontics to obtain maximum dental movements or to anchor a group of teeth during the multi-brackets fixed therapy of teeth. Commercial orthodontic micro-implants are formed from pure titanium, titanium alloys, and surgical stainless steel. An important respectable feature in implantology is to realize efficient osseointegration and to keep an intimate contact of tissue with the micro-implants by hindering bacterial multiplication [1]. 

There is ample literature that studied the long-term stability of implants, as they depend on the degree of inflammation that affects the soft tissues and causes loss of supporting bone, escalating progressive damage to the cortical bone and peri-implantitis [2,3]. Infection of the implants is frequently seen in a titanium-based prostheses; this can be prevented by decreasing the adhesion of microorganisms on the surface of implanted devices [4]. *Streptococcus* species, are the most predominant microorganisms that colonize implant surfaces and are responsible for many peri-implant and periodontal diseases [5]. 

Around the world, many different plant species have been used for oral treatments; one of the most significant plants is the Arak tree or Miswak that belongs to the family “*Salvadoraceae*” [6]. Aqueous extracts of *Salvadora persica* have been investigated as antibacterial, anti-caries, and anti-fungals that have been confirmed for oral hygiene [7]. They also contain several beneficial chemical constituents such as chloride (Cl^−^), fluoride (F^−^), sulfur, flavonoids, and resins [8].

The etal or metal oxides nanoparticles copper, silver, gold, CuO, ZnO, titanium dioxide (TiO_2_), or other non-metal ones as hydroxyapatite, silica (SiO_2_) and chitosan have been shown to provide the most effective antibacterial activity through interacting efficiently with microbial membranes [9,10]. Therefore, those nanoparticles can be used as a micro implant coating in orthodontics [11]. TiO_2_ and ZnO nanoparticles are widely used as antimicrobial coatings for micro-implants [12,13]. Recently nanoparticles have been studied successfully in dentistry in terms of diagnosis, the cure of oral cancer, and dentinal hypersensitivity. Nanofibers and nanoneedles have been biofilms’ wound dressings [14]. They also aid in preventing biofilm formation on the oral cavity surface; they can penetrate more deeply into microbial membranes [15,16]. 

Hydroxyapatite (HAP) is a material with many biomedical applications such as drug delivery, orthopedic dentistry, and antimicrobial uses. It has excellent biocompatibility, remarkable similarity of the composition of the bone mineral phase, and the ability to enhance the expression and osteoconductivity of cellular functions. HAP has a remarkable ability to penetrate and diffuse into the bone structure and support the growth of the bone [17]. HAP is extensively used in dentistry as a biomaterial for bone cavity fillings or metallic implant coatings. Also, it has been introduced in many fields due to its antimicrobial activities [18]. The synthetic hydroxyapatite ceramics are modified with small quantities of additives (CO_3_^−2^, F, Zn^+2^, Mg^+2^, and Mn^+2^ ions) to enhance the bioactive characteristic of the implants [19]. 

Chitosan is a versatile biopolymer with fiber, film, and micro/nanoparticle forming features due to its availability, low production cost, non-toxic nature, biocompatibility, biodegradability, and renewable nature [20]. It is a non-toxic natural polysaccharide that possesses a polycationic nature. Over the recent years, chitosan nanoparticles (CSNPs) have gained significant interest and are being used in a wide variety of different applications and products as well as for broad antimicrobial activities [21,22]. 

In this study, we prepared chitosan and HAP nanoparticles from *Salvadora persica* “Miswak” as a green biosynthesis of nanoparticles using plant extract as a reducing and stabilizer agent, then coated them on titanium micro-implants and evaluated their antimicrobial characteristics to be applied in decreasing the microbial aggregation around dental micro-implants.

## 2. Material & Methods

### 2.1. Ethical Approval

The plant study experimentation was performed in accordance with the International Guidelines for The Care and Use of Laboratory Animals and Plants. In addition, the ethical code followed the guidelines of the National Committee of Bioethics, King Abdulaziz City for Science and Technology, Saudi Arabia. 

### 2.2. Preparation of Salvadora persica “Miswak” Plant Extract

*Salvadora persica* (Miswak) chewing sticks were gathered from the twigs of Arak trees in Saudi Arabia (Jazan city). The leaves were identified in the Botany Department, Faculty of Science, Jazan University, Saudi Arabia. The department’s herbarium received a voucher with the specimen number JU/COP/19–7. Ten grams extract was filtered through a 0.45 mm filter after being chopped, dried, and boiled in 100 mL water. The aqueous leaf extraction had a final concentration of 0.01775 g/mL.

### 2.3. Synthesis of Chitosan Nanoparticles (CS) from Plant Extract by Ionic Gelation Method

Chitosan-polyacrylate lattices were prepared by the ionic gelation method in the Department of Polymers, National Research Centre, Giza, Egypt. The CS nanoparticles were obtained by stimulating a CS gelatin solution with Sodium tripolyphosphate (TPP).

*Salvadora persica* had been incorporated into TPP solution before the nanoparticles were formed. The extract was added drop-wise at three different concentrations (5%, 10%, and 20%) in relation to the amount of chitosan, and 1 percent Tween 80 was added in magnetic stirring for two hours followed by centrifugation solutions to prevent particle aggregation, and the chitosan solutions were then raised to pH 4.6–4.8 with 1N NaOH. A 0.1 percent sodium tripolyphosphate solution was made by dissolving 10mg of TPP in 10mL of deionized water and diluted to obtain different concentrations: 0.25, 0.50, 0.75, 1, 1.5, and 2 mg/mL. 

For eliminating any free plant extract, cellulose membrane (spectra/por-dialysis membrane; Mw 3000 Da) was utilized (non-encapsulated). After dialysis, the non-entrapped plant extract was recovered from the external aqueous phase and quantified using a calibration curve derived from extracted solutions of known concentration and measured using a spectrometer at 485 nm [23].
(1)$$\mathrm{Encapsulation}\;\mathrm{efficiency}\;\left(\%\right)=\left(\mathrm{total}\;\mathrm{amount}\;-\;\mathrm{free}\;\mathrm{amount}\right)\;/\mathrm{total}\;\mathrm{amount}\;\times 100$$
was used to calculate the percentage of encapsulation efficiency.

The clear solution, opalescent suspension, and aggregates were the three kinds of visually observed samples. The opalescent suspension zone corresponds to highly tiny particles. The chitosan particle suspension was centrifuged for 30 min at 12,000× *g*. In water, the pellet was resuspended. Before further use or analysis, the chitosan nanoparticles dispersion was freeze-dried.

Cellulose membrane\spectra; Mw 3000 Da was used for removing any free plant extract [24]. 

By measuring the absorbance rate at 485 nm with a spectrometer, the non-entrapped plant extract was recovered and measured using a calibration curve obtained from extracted solutions of known concentration.

### 2.4. Synthesis of Hydroxyapatite Nanoparticles from Salvadora persica “Miswak” Plant Extract 

Freshly cut *Salvadora persica* chewing sticks were dried and ground in 100 mL sterile distilled water was added to ten grams of *Miswak* and incubated for 48 h at 40 °C. The extract was centrifuged and filtered through a 0.45 mm filter paper. 1 M CaCl_2_ (Sigma-Aldrich, St. Louis, MO, USA) and 0.6 M sodium dihydrogen phosphate (Na_2_HPO_4_) (Sigma-Aldrich, St. Louis, MO, USA) were prepared at pH above 10 using 0.7 M NaOH (Sigma-Aldrich, St. Louis, MO, USA) solution. The prepared *Miswak* extract was used as a solvent, and then the CaCl_2_ solution was added. At the same time, the P:Ca ratio was maintained at 1.67, and the Na_2_HPO_4_ solution was added to obtain white precipitate that washed with deionized water [25,26]. 

### 2.5. Characterization of Chitosan and Hydroxyapatite Nanoparticles

The synthesized (CSNPs) were characterized through a UV-Vis spectrophotometer HITACHI U2300 (Chiyoda, Tokyo, Japan) range of absorbance from 200–600 nm. The morphologies and mean particle size of chitosan and hydroxyapatite nanoparticles were characterized using field emission scanning electron microscopy (FESEM). Elemental analysis of the prepared nanoparticles was carried out on an EDAX (FEG250, Quanta, Hillsboro, OR, USA), where transmission electron microscopy (TEM) was performed using JEO (Tokyo, Japan) for chitosan nanoparticles. A Nicolet Magna-IR Spectrometer (Magna 550; Nicolet, Madison, WI, USA), Fourier transform infrared spectroscopy analysis (FTIR) spectra were recorded in KBr pellets. CSNPs was used to identify and get an approximate idea of the possible biomolecules. Dry potassium bromide powder was compressed with the tested samples into a disc using a hydrostatic press.

### 2.6. Antimicrobial Activity of Chitosan and Hydroxyapatite Nanoparticles

#### 2.6.1. Bacterial Strains

Cultures of *Streptococcus sanguis, Streptococcus mutans*, *Streptococcus salivarius*, and *Enterococcus faecalis* were obtained from the faculty of dentistry, Jazan University, Saudi Arabia; the cultures were inoculated in an anaerobic chamber with 85% of N_2_, 10% H_2_, and 5% CO_2_ at 37 °C on sheep blood agar plates for 48 h. 

#### 2.6.2. Agar Well Diffusion Assay

The antibacterial activity of hydroxyapatite and chitosan nanoparticles was tested. Bacteria were incubated in an anaerobic jar in Luria–Bertani broth media (Himedia, Mumbai, India) for 24 h. A colony-forming unit (CFU) of each bacterium was adjusted to 0.5 McFarland standards (approximately 2–4 × 10^8^ CFU/mL), and wells were made in Muller-Hinton media (Himedia, Mumbai, India). Cultures of bacteria with positive and negative controls were inoculated in the plates. Different concentrations of nanoparticles (1.25, 2.5 & 5 mg/mL) were poured on marked wells and incubated at 37 °C for 24 h. The zone of inhibition was measured in mm [27].

#### 2.6.3. Minimum Inhibitory Concentration (MIC) and Minimum Bacterial Concentration (MBC)

The MIC and MBC of Hydroxyapatite and chitosan nanoparticles were determined by the broth microdilution method. 100 µ of 0.5 McFarland’s bacterial suspension represents 10^8^ cells/mL was inoculated into each well of microtiter plates. Then a tenfold serial dilution (0, 0.25, 0.5, 1, 2, 4, 8, 16, 32, 64, 128, and 256 μg/mL) of the nanoparticles was added and incubated for 24 h at 37 °C [28].

#### 2.6.4. Biofilm Formation on the Surfaces of Hydroxyapatite and Chitosan Nanoparticles

Two mL brain heart infusion (BHI) broth (Himedia, Mumbai, India) were added to a 24-wells plate, and sterile disks prepared for testing with or without aging were placed into the wells. Twenty-five µL of the diluted, mixed species *of*
*E. faecalis*, *S. mutans* bacteria was treated with hydroxyapatite, and a chitosan nanoparticle suspension was added to each well and incubated in anaerobic condition for 24 h; the planktonic bacteria and biofilm formation on the disk in the culture medium were then assessed [29].To start the biofilm formation assay, precultures of each bacterium stored at −80 °C were grown in 10 mL of brain heart infusion (BHI) medium (Difco Laboratories, Detroit, MI, USA) for 24 h at 37 °C to full growth. To evaluate biofilm formation in cocultures of *E. faecalis*, spp. and oral *S. mutans* spp., 20 μL of an enterococcal cell suspension and 20 μL of the other bacterial cell suspension were mixed in a well of a 96-well (flat-bottom) microtiter plate, along with 160 μL of tryptic soy broth (without dextrose, supplemented with 0.25% sucrose) after coating the plates with 25 µL of the diluted, mixed species *of E. faecalis, S. mutans* bacteria treated with hydroxyapatite, and chitosan nanoparticles suspension for 30 min at 37 °C. To evaluate biofilm formation by single cultures, 20 μL of bacterial cell suspension and 180 μL of TSBS were added to each well. The plates were incubated at 37 °C for 16 h under anaerobic conditions, and the liquid medium was removed. The wells were rinsed a second time with d-water, air-dried, and stained with 0.25% safranin for 15 min. After staining, the plates were rinsed with d-water to remove excess dye and then air-dried. The biofilm mass was dissolved with ethanol, and the stained biofilm was quantified by measuring the absorbance at 492 nm using a microplate reader. The pH of the supernatant was determined before and after culturing for 16 h. Live/dead staining for visualization of mixed species of *E. faecalis, S. mutans* viability was examined by a confocal laser scanning microscope (CLSM) (FluoView FV1000, Olympus, Tokyo, Japan) [30].

The discs coated with biofilms of mixed species of *E. faecalis, S. mutans* were washed three times with sterile saline, and then the remaining bacteria were stained with the fluorescent dyes using the Live/Dead BacLight Bacterial Viability Kit. Acridine Orange fluoresces green (AO) (Sigma-Aldrich, St. Louis, MO, USA) was added to stain live cells while ethidium bromide fluoresces red (EB) (Sigma-Aldrich, St. Louis, MO, USA) was added to stain dead cells with a 15 min incubation in the dark at room temperature [31].

The samples were washed gently with distilled water and examined by CLSM. The green fluorescence emission of live bacteria was estimated through excitation with a 488 nm laser. In contrast, the red fluorescence emission of dead bacteria with damaged membrane was estimated through excitation with a 543 nm [32].

#### 2.6.5. Biopolymer Layer (HAP/Chitosan) Formation over the Titanium Micro-Implants Surface 

Micro-implants were rinsed in ethanol by ultrasonic radiation and then dried. Micro-implants were dropped into the biopolymer (HAP/chitosan) solution from the acrylic surface and stirred continuously for two hours, and then dried. 

#### 2.6.6. Preparation of Hydroxyapatite/Chitosan-Polyacrylate Lattices by In Situ Emulsion Polymerization Technique

Twenty mL of chitosan-polyacrylate emulsion lattices was mixed with 20 mL of hydroxyapatite nanoparticles. A0.2 mL of 25% glutaraldehyde aqueous solution was added to this mixture and allowed to proceed for 15 min under sonication (350 Hz) and continued for 2 h at 50 °C using magnetic stirring. The solution was cooled and filtered.

#### 2.6.7. Surface Characterization of Titanium Micro-Implants with Regular HAP and Chitosan Nanoparticles

Micro Titanium implants TI-6AI-4V Vector TAS (Ormco Corporation Orange, CA, USA) were rinsed in ethanol by ultrasonic radiation and then dried; different HAP nanoparticle concentrations were applied (0.2, 0.5, 1, 2, 4, 6 mg/mL) for 3 min, and then rinsed with deionized water for five seconds, and dried in an oven at 55 °C for 24 h; the samples were cleaned with deionized water and referred as HAP nanoparticle coating. For chitosan nanoparticle loading, the HAP coating was soaked into the 2% acetic acid solutions with different Chitosan nanoparticle concentrations (0.2, 0.5, 1, 2, 4, 6 mg/mL) for 3 min, and then rinsed with deionized water for five seconds, and dried in an oven at 55 °C for 24 h. Four groups of titanium micro-implants were prepared from (A–D) and coated with HAP and chitosan nanoparticles, and the surface was characterized by scanning electron microscopy (SEM).The first group (A) represented non treated titanium micro-implants surface (control), the second group (B) were micro-implants Ti- surface coated with hydroxyapatite nanoparticles at different magnifications, and the third group (C) were micro-implants that were Ti- surface coated with chitosan nanoparticles at different magnification. Finally, the fourth group (D) were micro-implants that were Ti- surface coated with hydroxyapatite nanoparticles and chitosan at different magnifications.

## 3. Results

### 3.1. Characterization of Chitosan Nanoparticles

The synthesized chitosan nanoparticles were characterized by ultraviolet-visible spectroscopy (UV-Vis); the formed chitosan nanoparticles in the solution were scanned and detected in the range of 200–750 nm in a spectrophotometer to verify the formation of nanoparticles that showed the peak formed at 228 nm as seen in Figure 1. The morphology of chitosan nanoparticles powder was determined in the scanning electron microscopy (SEM) chamber at the acceleration voltage of 20 KV. Figure 2 showed SEM micrographs of powdered chitosan as a uniform crystalline surface with dense structure, small voids, and many dimples, which indicated the presence of pores. Figure 3 represents the quantitative elemental composition and the EDX spectra of the samples. The results showed the presence of C, N, and O in the chitosan. The TEM, as shown in Figure 4, with the particle size was observed around 70 to 100 nm.

#### Fourier Transform Infrared Spectroscopy Analysis

Fourier transform infrared spectroscopy analysis (FTIR) of (CSNPs) is shown in Figure 5; it is clear that FTIR showed the intense and wide band at 3220 cm^−1^ related to the vibration of -OH group, the characteristic peaks at 1513 cm^−1^ and 1040 cm^−1^ are due to the vibration of C=O.

### 3.2. Characterization of Hydroxyapatite Nanoparticles (HAPNPs)

#### Scanning Electronic Microscope (SEM) Analysis

The SEM images of synthesized HAPNPs are shown in Figure 6, where formed agglomerated and/or porous molecules can be seen.

The chemical element formulation of produced HAPNPs is shown in Figure 7, and showed the amount of phosphorus and calcium. The atomic and weight percentages are also presented. The EDX result showed a P/Ca ratio around 1.69, which was below two. The ideal P/Ca ratio of Hydroxyapatite is 1.66. The structure and morphology of the HAPNPs samples were confirmed by the TEM, as shown in Figure 8, with a particle size of 50 nm observed.

The functional groups of the HAPNPs were analyzed and identified by FTIR Perkin Elmer, as seen in Figure 9. The graph showed broad bands around 1644.47 cm^−1^ and 3448.02 cm^−1^.

### 3.3. Antimicrobial Activity of Chitosan, Hydroxyapatite Nanoparticles

#### Measurement of Inhibition Zone Diameter

The measurements of the inhibition zone diameter of chitosan and hydroxyapatite nanoparticles for *S. salivarius*, *S. mutans*, *S. sanguis*, and *E. faecalis* are illustrated in Table 1. Both nanoparticles’ greatest zones of inhibition were found at a concentration of 10 mg/mL and a decrease in the zone of growth inhibition with the reduction of concentration, as seen in Figure 10. The lowest zones of inhibition by those nanoparticles against bacteria were found at a concentration of 2.5 mg/mL. The diameter of inhibition zone of HAPNPs against bacteria was more than that of Chitosan with all the above-mentioned bacterial species, especially for *S. mutans* bacteria that was showed maximum zone of inhibition (18.3 mm) at the highest concentration of HAPNPs 5 mg/mL and (12.8 mm) at the lowest concentration of HAPNPs 2.5 mg/mL.

The results of the MIC value of CSNPs against testedoral bacteria are seen in Table 2, Figure 11. The MIC values of CSNPs were 8 μg/mL on *E. faecalis*, *S. mutans* and *S. salivarius*, with St. deviation ± 0.07, 0.014 respectively, while this was 16 μg/mL on *S. sanguis* with St. deviation ± 0.014. At the same time, the MBC of CSNPs was evaluated, as presented in Table 3. The same values were found for *S. salivarius, S. mutans* and *E. faecalis* (4 μg/mL) with St. deviation ± 0.07, 0.28 respectively, while the MBC of CSNPs should be *S. sanguis* was 8 μg/mL with St. deviation ± 0.014.

The MIC values of HAPNPs as seen in Table 3 and Figure 12 were 16 μg/mL for *E. faecalis*, and *S. sanguis* with St. deviation ± 0.07, 0.014, 8 μg/mL with St. deviation ± 0.07 for *S. salivarius* and 4 μg/mL with St. deviation ± 0.07 for *S. mutans*. On the other hand, MBC of HAPNPs on *S. mutans* was the same value of MIC (4 μg/mL) with St. deviation ± 0.14 that indicate the bacteriostatic and bactericidal effect of HAPNPs while MBC of HAPNPs on the other three bacteria was varied; it was 8 μg/mL *E. faecalis* and *S. sanguis* with St. deviation ± 0.14, 0.28 and 4 μg/mL for *S.salivarius* with St. deviation ± 0.07.

Confocal Laser Scanning Microscopy (CLSM) was used to investigate the antibacterial activity of hydroxyapatite and chitosan nanoparticles on biofilm of mixed species of *Enterococcus faecalis and Streptococcus mutans* bacteria as seen in (Figure 13 and Figure 14) based on membrane integrity. The fluorescence image of the control, as seen in (Figure 13 and Figure 14A), showed several intact viable green-colored cells. On the contrary, *Enterococcus faecalis and Streptococcus mutans* biofilms cells were treated with hydroxyapatite nanoparticles as seen in (Figure 13B), where they showed numerous strong red colors with several less distinctive red cells of damaged boundary layers. On the other hand (Figure 14B) *Enterococcus faecalis and Streptococcus mutans* biofilms cells treated with chitosan nanoparticles showed less strong red color cells with less distinctive red cells of damaged boundary layers.

### 3.4. Surface Characterization of the Titanium Micro-Implants Coated by Hydroxyapatite and Chitosan Nanoparticle

The SEM images of the titanium micro-implants (four groups from A–D); the first group (A) were (control), the second group (B) were micro-implants Ti- surface coated with hydroxyapatite nanoparticles, the third group (C) were micro-implants Ti- surface coated with chitosan nanoparticles, and the fourth group (D) were micro-implants Ti- surface coated with hydroxyapatite nanoparticles and chitosan, showing various thickness of a disperse coating of nanoparticles over the surface of titanium micro-implants and exhibited a mesoporus/macroporous surface matrix with a pore of roughly diameter 16 to 60 nm at (Figure 15).

## 4. Discussion

The foundation of nanoparticles with various biomaterials to increase their antimicrobial features was extensively investigated in our study. 

The characterization of prepared CSNPs by UV-Vis showed, as seen in Figure 1, the peak at 228 nm; this could be as a result of the amido group present in Chitosan [33,34]. Similarly, a peak was observed at 201 by Liu et al. [35], where they prepared and characterized nanoparticles from trypsin based hydrophobically-modified chitosan. Also, Megha et al [33] demonstrated a peak for CSNPs at 223 nm when prepared with CSNPs and studied their in-vitro characterization. SEM analysis recorded that the size of the CSNPs ranges from 70 to 100 nm. 

Morphologically, in our result, the prepared CSNPs were agglomerated with small spheres with porous structures, and this result agreed with Zhao et al., 2021 [36]. FTIR spectra of (CSNPs) were shown in Figure 5; CS characteristic peaks are amide groups whose presence of the residual N-acetyl group were confirmed by the bands at around 1513 cm^−1^ (C=O stretching of amide I) and 1350 cm^−1^ (C–N stretching of amide III), respectively. An absorption band at 3220 cm^−1^ indicates O–H stretching. The major absorption band between 1040 and 1090 cm^−1^ represents the anti-symmetric stretching of the C–O–C bridge; however, the skeletal vibrations involving the C–O stretching are characteristic of its saccharide structure

Fourier transform infrared spectroscopy analysis (FTIR)of CSNPs were shown in Figure 5; CS characteristic peaks are the amide groups’ presence in the residual N-acetyl group confirmed by the bands at around 1513 cm^−1^ (C=O stretching of amide I) and 1350 cm^−1^ (C–N stretching of amide III), respectively. An absorption band at 3220 cm^−^^1^ indicates O–H stretching. The major absorption band between 1040 and 1090 cm^−1^ represents the anti-symmetric stretching of the C–O–C bridge; however, the skeletal vibrations involving the C–O stretching are characteristic of its saccharide structure [37,38,39,40,41]

The orphology of the HAPNPs was observed in SEM images, as seen in Figure 6. Some particles were agglomerated and porous with irregular sHAPes that considered it advantageous to increase the permeability of the tissue on implants inside the body, where it can be used as a biomaterial; improving interaction between the biological environments and implant [42]. The results were found to be harmonized with the data of Ferraz et al. [43]. The presented spectrum of EDX analysis, as shown in Figure 7, assessed the Ca/P value of the synthesized HAPNPs, which was 1.67, which was near to the Ca/P ratio of the human bone [44]. The FTIR spectrum was scanned, as shown in Figure 9, from 400–4000 cm^−1^. FTIR spectrum showed the characteristic absorption peaks of HAPNPs attributable to sharp bending of the (OH) groupband at 3568 cm^−1^, the characteristic band at 570,1041 represented (PO_4_)^−3^, and also the asymmetric P–O stretching vibration on the PO_4_ bands at 869 cm^−1^ as a distinguishable peak and the sharp peaks at 602,570 cm ^−1^ corresponding to the triply degenerate bending vibration of PO_4_ in hydroxyiapatite. Characteristic peaks of HAP are found at 869 cm^−1^ and 1461 cm^−1^, which represents the elimination of the (CO_3_)^−2^ group due to the calcification of hydroxyapatite at a high temperature of 900 °C [45]; our results were concerned with [46]. Many studies recorded that the ions of nanoparticles play an important role in the inactivation of bacteria by reacting with SH groups of proteins [47]. Also, oxidative phosphorylation from uncoupling respiratory electron transport interferes with membrane permeability to protons and inhibits respiratory chain enzymes. Others reported that nanoparticles destroy the DNA replication, the production of reactive oxygen species, and the formation of ATP, finally damaging cell membranes and causing the apoptosis of the bacterial cells [48]. 

Interestingly, the novel hydroxyapatite and chitosan nanoparticles in the current study demonstrates promising antimicrobial effects versus both biofilm and planktonic bacteria. Hence, these nanoparticles are greatly effective against tested oral pathogens such as *E. faecalis*, *S. mutans*, *S. sanguis*, and *S. salivarius* at 5–1.25 mg/mL for their planktonic modes, as illustrated in Table 1. The greatest zones of inhibition by chitosan and Hydroxyapatite nanoparticles were found at a 5 mg/mL concentration and a decrease in the zone of growth inhibition with the reduction in concentration. HAPNPs have more significant bacterial activity than chitosan, especially against *S. mutans* bacteria that showed a maximum zone of inhibition of 18.3 mm at the highest concentration of HAPNPs (5 mg/mL) and (12.8) mm at the lowest concentration (1.25) mg/mL. The estimation of MIC was confirmed and used for the comparative testing of new materials [49].

This was the first study in the literature to include MIC and MBC of chitosan and hydroxyapatite nanoparticles against pathogenic oral bacteria, as seen in Table 2 and Table 3. The results proved that the MIC values are varied according to the bacterial strains; it was recorded as 16, 4, 16, 8 µg/mL for HAPNPs and 8, 8, 16, 8 µg/mL, for CSNPs, while the MBC values recorded 8, 4, 8, 4 µg/mL for HAPNPs and 4, 4, 8, 4 µg/mL for CSNPs. 

In contrast to our results, previous studies concluded that the antimicrobial activity of HAPNPs has potential antimicrobial activity against Gram-negative than Gram-positive microbes [50,51]. 

CLSM was used to evaluate bacterial viability and anti-biofilm efficacy [52]. Consequently, about bacterial viability, the ratio of red to green fluorescent cells was indicated as a quantitative index. The obtained results of CLSM imaging of mixed-species *E. faecalis* and *S. mutans* (treated with both nanoparticle) came in accordance with the MIC data as seen in (Figure 13 and Figure 14). The fluorescence image of the control (Figure 13A) showed several intact viable green-colored cells. On the contrary, mixed bacteria species treated with HAPNPs (Figure 13B) showed numerous strong red colors with several less distinctive red cells of the damaged boundary; on the other hand, the bacterial mixture treated with CSNPs showed less potent red color cells (Figure 14B). However, previous studies have stated the significant bactericidal activity of hydroxyapatite and chitosan nanoparticles [53,54]. 

Prevention of implant failures is one of the main important factors for induction of surface modification of ti-based micro implants to fulfill successful functional treatment results; bacterial colonization results in inflammation of soft tissue [55,56]. Therefore, nanoparticles have been applied to decrease the bacterial colonization and increase the bone incorporation of dental micro-implants [57,58]. 

In our study, titanium micro-implants coated by hydroxyapatite and chitosan nanoparticles showed smaller noble metal deposits dispersed over an irregular surface topography (Figure 15A–D). The deposits varied in size from approximately 16 to 60 nm in diameter and were not distributed uniformly along the surface of micro implant. Coinciding with our data, nanoparticles increase the TiO_2_ implant stability and density of osteoblast cells on the implant rather than reducing bacterial adhesion [59,60].

## 5. Conclusion

This study demonstrated the successful significant antibacterial activity of prepared green synthesized hydroxyapatite and chitosan nanoparticles. Also, the study has shed light on the possible application of nanoparticles for the efficient coating of titanium micro-implants with significant potential for their use in facing different challenges in dentistry. 

## Figures and Tables

**Figure 1 microorganisms-10-00581-f001:**
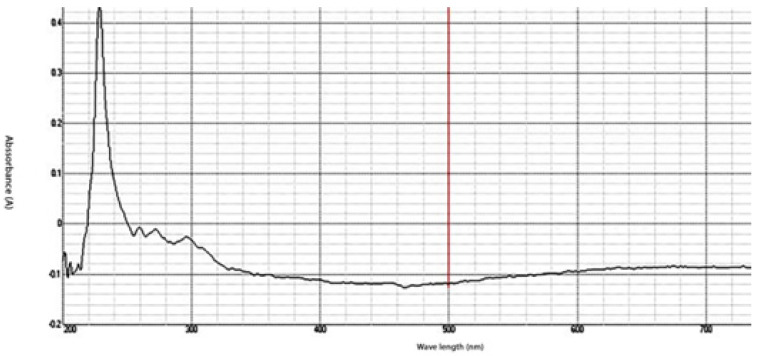
UV-Analysis- The absorption peak for Chitosan nanoparticles (CSNPs) in the range of 200–750 nm.

**Figure 2 microorganisms-10-00581-f002:**
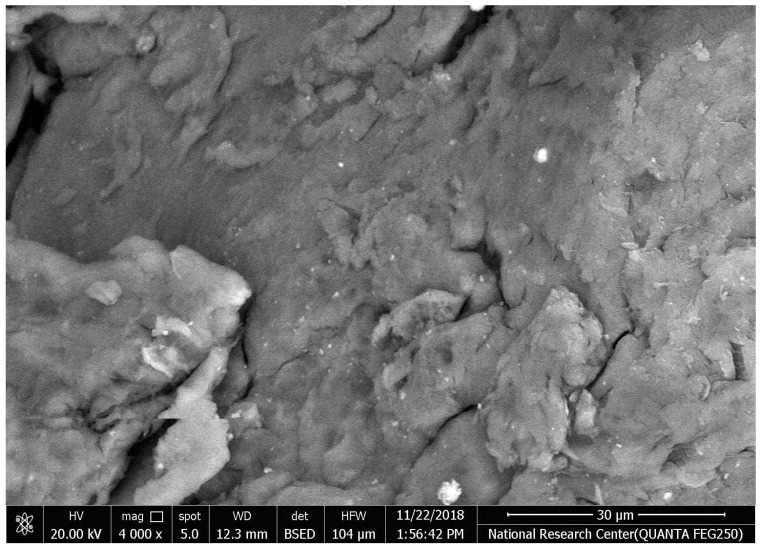
The morphology of chitosan nanoparticles powder by Scanning electron microscopy (SEM) image of chitosan.

**Figure 3 microorganisms-10-00581-f003:**
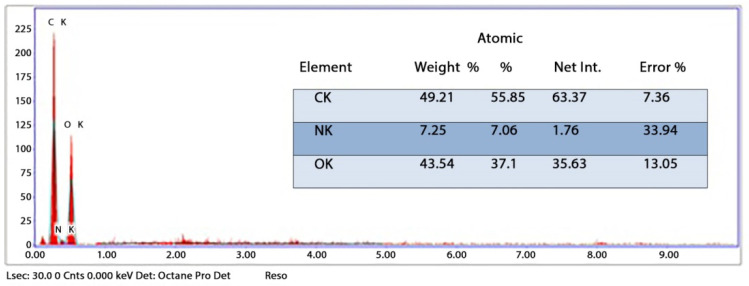
Energy-dispersive X-ray spectroscopy (EDX) analysis of CSNPs powder.

**Figure 4 microorganisms-10-00581-f004:**
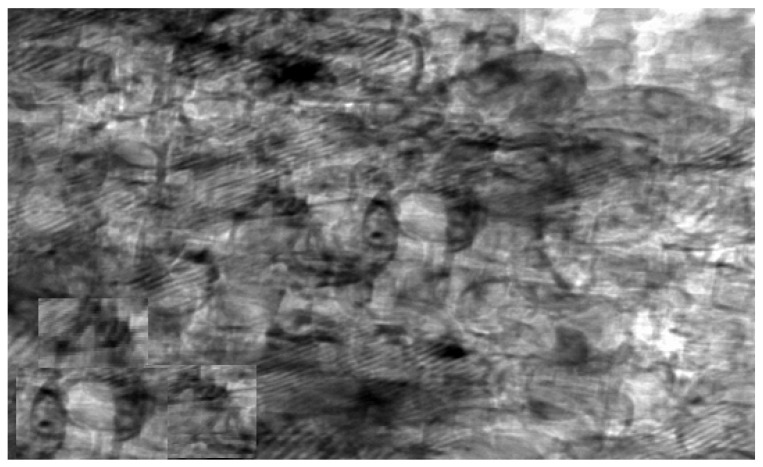
Transmission electron microscope (TEM) image of chitosan nanoparticles (CSNPs).

**Figure 5 microorganisms-10-00581-f005:**
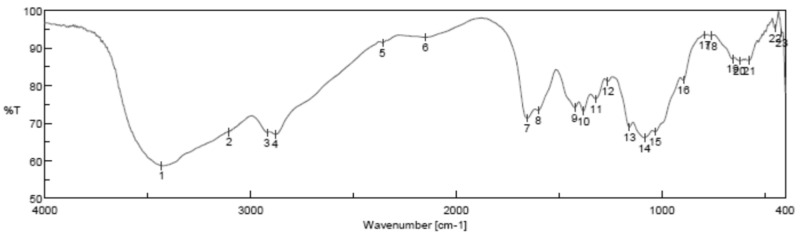
Fourier transform infrared spectroscopy analysis (FTIR) of chitosan nanoparticles (CSNPs) with various components.

**Figure 6 microorganisms-10-00581-f006:**
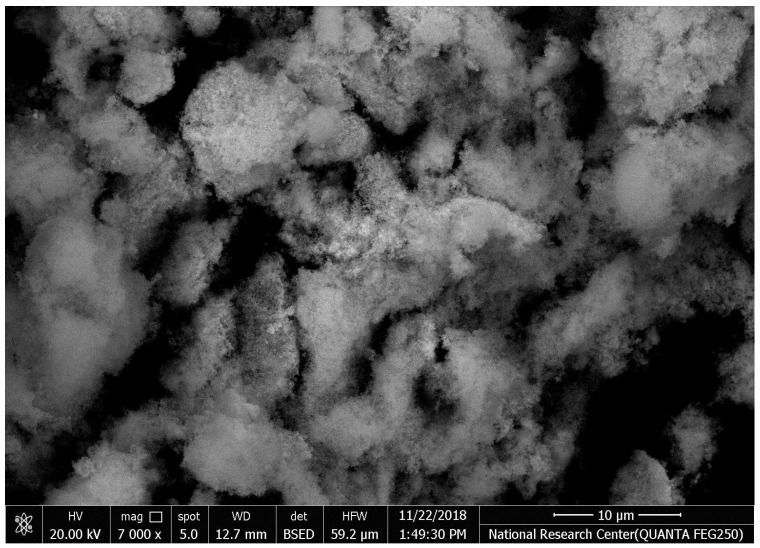
Scanning electron microscopy of hydroxyapatite nanoparticles (HAPNPs) with homogeneity 100 k zoom (energy 2.00 Kv).

**Figure 7 microorganisms-10-00581-f007:**
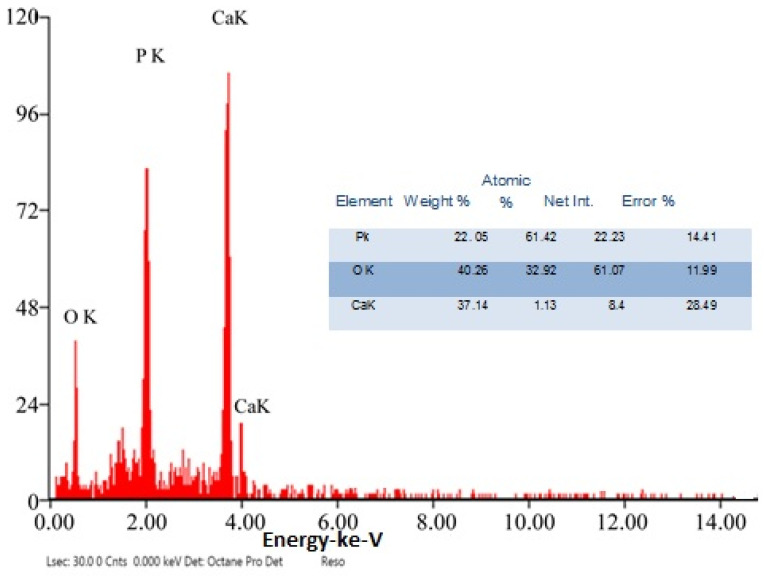
EDX analysis of HAPNPs powder.

**Figure 8 microorganisms-10-00581-f008:**
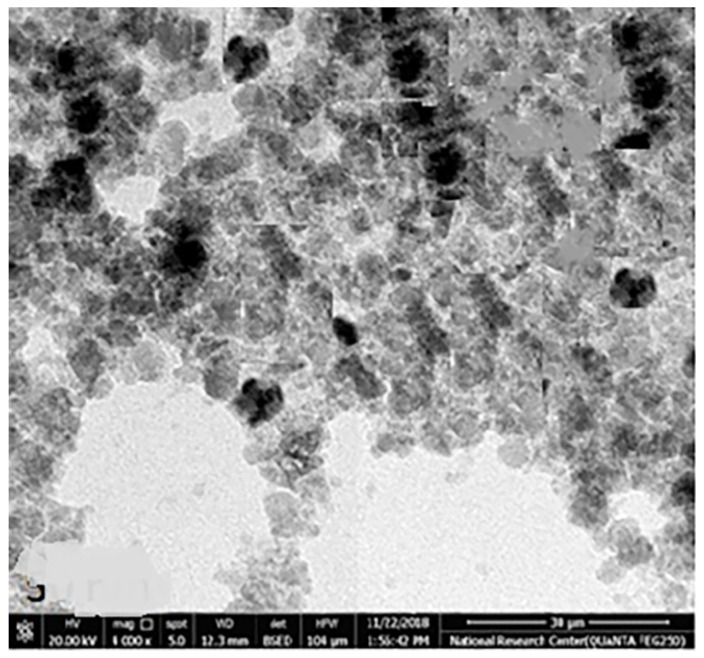
Transmission electron microscopy of HAPNPs.

**Figure 9 microorganisms-10-00581-f009:**
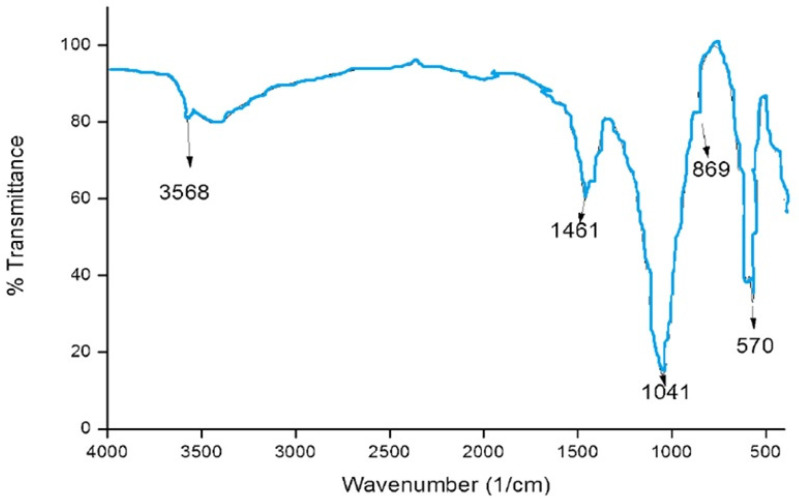
FTIR analysis of HAPNPs powder.

**Figure 10 microorganisms-10-00581-f010:**
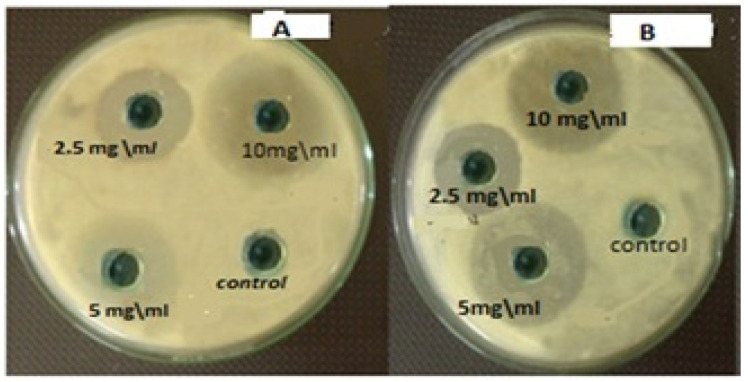
Antibacterial activity of different concentrations of HAPNPs and chitosan NPs against oral mouth bacteria. (**A**) Antibacterial activity of different concentrations of HAPNPs against *Streptococcus mutans*, (**B**) Antibacterial activity of different concentrations of chitosan NPs against *Streptococcus mutans*.

**Figure 11 microorganisms-10-00581-f011:**
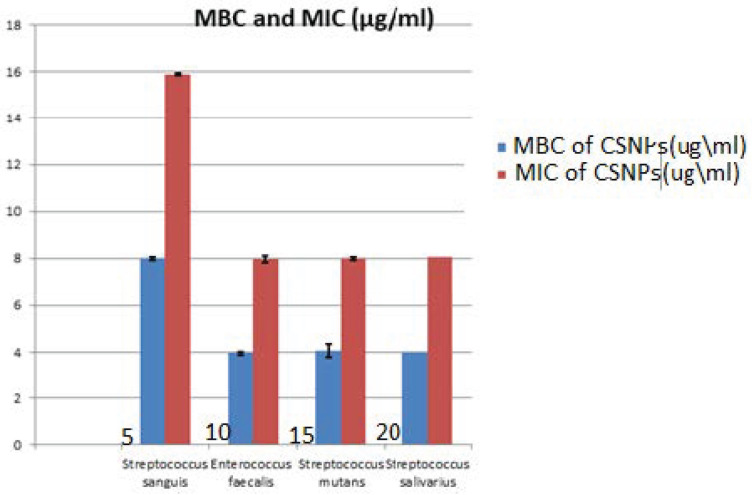
Minimum Inhibitory Concentration (MIC) and Minimum Bacterial Concentration (MBC). of chitosan nanoparticles (CSNPs) against *Streptococcus sanguis*, *Enterococcus faecalis*, *Streptococcus mutans*, and *Streptococcus salivarius*.

**Figure 12 microorganisms-10-00581-f012:**
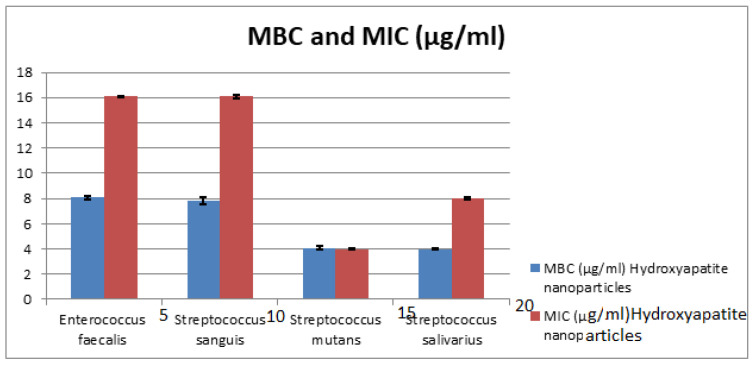
Minimum Inhibitory Concentration (MIC) and Minimum Bacterial Concentration (MBC) of CSNPs. HAPNPs against *Streptococcus sanguis, Enterococcus faecalis, Streptococcus mutans, and Streptococcus salivarius*.

**Figure 13 microorganisms-10-00581-f013:**
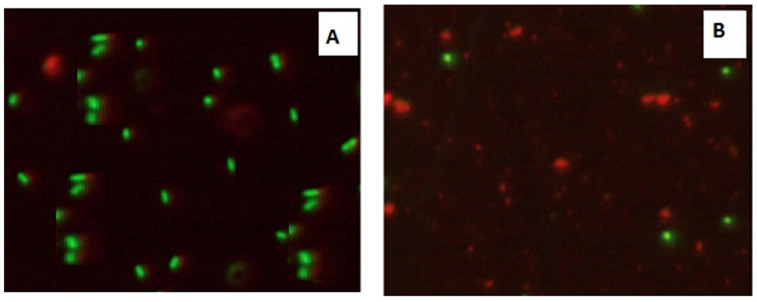
Confocal Scanning Laser Microscopy (CSLM) of mixed species *Enterococcus faecalis* and *Streptococcus mutans*; before (**A**) and after treatment (**B**) with CSNPs.

**Figure 14 microorganisms-10-00581-f014:**
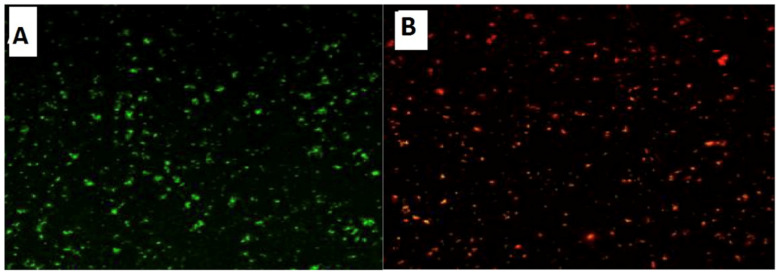
Confocal Scanning Laser Microscopy (CSLM) of mixed species *Enterococcus faecalis* and *Streptococcus mutans*; before (**A**) and after treatment (**B**) with HAPNPs.

**Figure 15 microorganisms-10-00581-f015:**
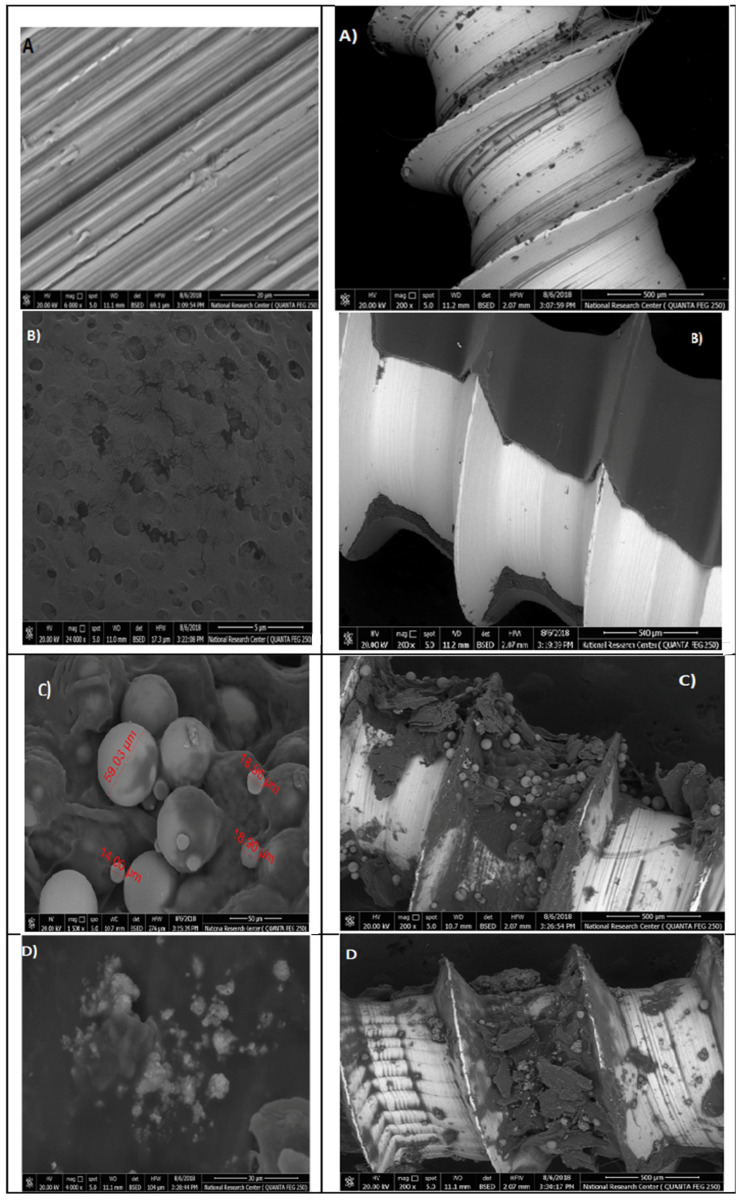
Surface characterization of the titanium micro-implants scanning electron micrographs showed: Group (**A**) titanium micro-implants surface (control); Group (**B**) micro-implants Ti- surface coated with HAPNPs; Group (**C**) micro-implants Ti- surface coated with CSNPs; Group (**D**) micro-implants Ti- surface coated with HAPNPs and CSNPs.

**Table 1 microorganisms-10-00581-t001:** The diameters of inhibition zones in mm of chitosan and hydroxyapatite nanoparticles against oral microorganisms.

	*Streptococcus salivarius*		*Streptococcus mutans*		*Enterococcus faecalis*		*Streptococcus sanguis* mm	
Concentration mg/mL	Chitosan NPs	Hydroxyapatite NPs	Chitosan NPs	Hydroxyapatite NPs	Chitosan NPs	Hydroxyapatite NPs	Chitosan NPs	Hydroxyapatite NPs
	Inhibition zones in mm							
10	13	15	16.2	18.3	12.4	14.5	14.8	16.09
5	11.32	13.43	13.20	14.43	10.3	11.3	11.8	13.70
2.5	10	9.07	11	12.08	8.6	9.3	9.4	10

NPs = nanoparticles.

**Table 2 microorganisms-10-00581-t002:** Minimum Inhibitory Concentration (MIC) and Minimum Bacterial Concentration (MBC) of CSNPs.

Bacteria	MBC (µg/mL)		Average	St. Deviation	MIC(µg/mL)		Average	St. Deviation
	Chitosan Nanoparticles				Chitosan Nanoparticles			
*Streptococcus sanguis*	8.1	7.9	8	0.14	16	15.8	16	0.14
*Enterococcus faecalis*	4	3.9	4	0.07	8	7.9	8	0.07
*Streptococcus mutans*	4.1	4	4	0.07	8.1	7.9	8	0.14
*Streptococcus salivarius*	4.2	3.8	4	0.28	8.1	8	8	0.07

**Table 3 microorganisms-10-00581-t003:** Minimum Inhibitory Concentration (MIC) and Minimum Bacterial Concentration (MBC) of HAPNPs.

Bacteria	MBC (µg/mL)		Average	St. Deviation	MIC (µ/mL)		Average	St. Deviation
	Hydroxyapatite Nanoparticles				Hydroxyapatite Nanoparticles			
*Enterococcus faecalis*	8.1	7.9	8	0.14	16.1	16	16	0.07
*Streptococcus sanguis*	7.8	8.2	8	0.28	16.1	15.9	16	0.14
*Streptococcus mutans*	4.1	3.9	4	0.14	4	3.9	4	0.07
*Streptococcus salivarius*	4	3.9	4	0.07	8	7.9	8	0.07

## Data Availability

Data will be provided upon request.

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
