# Peer review of "Incorporation of Plant Extracted Hydroxyapatite and Chitosan Nanoparticles on the Surface of Orthodontic Micro-Implants: An In-Vitro Antibacterial Study"

_microorganisms, 2022, doi:10.3390/microorganisms10030581_

Round 1

Reviewer 1 Report

This is an interesting work studying materials capable to reduce damage arising from interactions of orthodontic micro-implants and four oral pathogenic bacteria.

1) There is no consistency in using capitals in the title of the manuscript - I would suggest:  Incorporation of Plant Extracted Hydroxyapatite and Chitosan Nanoparticles on the Surface of Orthodontic Micro-implants: An  in-Vitro Antibacterial Study

2) L52-53 Metal or metal oxides nanoparticles; “copper, silver, gold, CuO, ZnO, titanium dioxide (TiO2), hydroxyapatite, silica (SiO2) and chitosan” have been shown the most effective antibacterial activity through interacting efficiently with microbial membranes [9,10].  Hydroxyapatite, silica, and chitosan are neither metals nor metal oxides nanoparticles, but yes, they show antibacterial properties.

3) L67-69 The synthetic hydroxyapatite ceramics are modified with small quantities of additives (CO3-2, F, Zn+2, Mg+2 and Mn+2 ions) to enhance the bioactive characteristic of the implants [19]. I read ref [19] but could not find any mention related to the HAp modifications. HAp NPs were synthesised, their structure and morphology analysed and antimicrobial activities related to gram positive and gram negative bacteria were examined.

4) L109 -111 The percentage of encapsulation efficiency was defined as 'Encapsulation efficiency (%)= (total amount – free amount/total amount) ×100'. The formula should be written as Encapsulation efficiency (%)= (total amount – free amount) /total amount ×100.

5) L193-194 Micro implants were dropped into the biopolymer (HA/chitosan) solution from the acrylic surface and stir continuously for two hours and dried. Please explain.

6) L198 First group (A) represented naïve titanium microimplants surface. Please explain.

7) L 219-219 Figure 1. The conclusion about the CS particle size seems to be far-fetched and ill-founded. The authors suggest that the peak in UV-VIS spectra situated at 228nm (Figure1) is related to the CSNP size 70 to 100nm and argue that ref [31] and [32] show peaks for chitosan in the UV region. In reality, the chitosan peak in ref [31] arises from silver nanoparticles produced from silver nitrate. They use 2 plant extracts, chitin and chitosan. The maximum in Fig. 6 ref [31] is at 310nm but it is for a chitosan biocomposite. In ref [32], there is no UV-VIS spectra at all. The authors S.B. Aziz, O.G. Abdullah and D.R. Saber in the paper Investigation of metallic Silver nanoparticles through UV-VIS and Optical Micrograph Techniques (Int. J. Electrochem. Sci 12, 2017,363-373 doi. 10.20964/2017.01.22) present a UV-VIS spectra for pure chitosan showing a peak at ca 215nm. Taking into account all information provided in the manuscript one cannot conclude that CSNPs were produced. 

8) L 237 Figure 6 shows that most HApNP particles are aglomerated and/or porous. Also, there is no bar visible on the left side of Figure 8.

9) L 269 Figure 10. The caption is insufficient.

10) L 281, L 293  Figures 11,12 The manuscript text does not correspond to the caption.

11) The interaction of HApNPs and CSNPs with biofilms was observed and evaluated qualitatively, using a Confocal Laser Scanning Microscope. L 297: There is missing information: how were the biofilms prepared? Tables 1 to 3 show results for 4 different strains of bacteria operating in planktonic mode.

12) L 320-321 Figure 15 Unsatisfactory caption. The figure shows SEM images of the surface (not surface characterization) of titanium micro-implants. The magnification is roughly the same in all cases but the coating materials are different.  L 397 How did the noble metal deposits materialize on the titanium micro-implants? How was their size (16-60nm) estimated/measured?

Reviewer 2 Report

Report attached

Reviewer 3 Report

In this study the authors sought to study the structural and morphological applications of hydroxyapatite and chitosan nanoparticles for coating of microimplants for the purpose of combating oral pathogenic bacteria.  The authors analyzed the crystal morphology, phase composition, particle size and surface functional groups of the nanosamples. The authors observed that hydroxyapatite nanoparticles has increased antibacterial activity and bacterial cell damage vs chitosan nanoparticles. 

Although an interesting in vitro study, several issues must be addressed.

In the abstract, when using the abbreviation HAP for the first time please explain what it stands for.

The introduction could be more to the point and should focus more on the matter at hand.

What is the null hypothesis of the study?

The resolution on fig 1,3,4,5,8 should be improved

Line 330: capitalize the name Priya

What are the limitations of the study?

Round 2

Reviewer 1 Report

1) Abstract has to be changed. Remove "In this study". "Chitosan nanoparticles showed (MICs) of 8 μg ml-1 and 16 μg ml-1 25 for (Streptococcus salivarius, Streptococcus mutans, Enterococcus faecalis, and Streptococcus sanguinis) respectively," There are 2 values of MICs (remove brackets) and 4 bacteria strains, i.e. it is impossible to guess, which value of MIC corresponds to which bacteria strain. Similarly, for HAP you have 3 values of MICs and 4 bacteria strains.

2) Unify the terminology and abbreviations used throughout the manuscript. You use Hydroxyapatite (lines 151,181 etc.), Hydroxyl apatite( line 386) and Hydroxy Appetite (lines 162, 174, 239 etc.) and abbreviations HA, HAP and HAp. You use both micro implants and micro-implants etc..

3) Infection of the implants is frequently seen in titanium-based prosthesis. (lines 45-46) (should be plural prostheses)

4) There are two Figures 1. The caption should be changed to reflect the fact that the UV region is below 380nm and the visible region is from 380nm to 750nm. Lettering (y-axis) has to be turned by 180 degrees. There are also two Figures 3,4,5,8,11,12. Captions to all Figures should be re-written.

5) Some explanations (for example lines 251-257) need to be re-written in sentences. The explanation I requested was provided but not in the form of concise sentences. It looks as if notes were written to be turned into sentences but this did not happen.

6) There are no units shown on y-axis of Figures 11 and 12.

7) CFU should be decoded when used for the fist time (line 179) as Colony Forming Unit

8) Captions of Figures 1,2,4,5 and 6 should be re-written

9) Line 313 should be corrected 3.2. “Charac HApNPs' of hydroxyapatite nanoparticles (HApNPs)” Did the authors mean Characterization of hydroxyapatite nanoparticles ?

10) Line 315 typo “demonestrated”  (consider demonstrated)

11) Figure 2 (line 297) Why is it there at all? What does it show? At that magnification, no conclusion can be made about the nature or the size of chitosan particles

12) Figure 4 (line 301) the same comment applied

13) Line 304 typo Fourier transform (not “transforms”) infrared spectroscopy

14) Captions to Figure 8 – The same text is written twice (line 329 and 332). Also, the 30 mm bar is about 3cm long. The authors claim that the particle size is 50nm, in which case it would be depicted as 0.05mm. Please explain.

15) Line 339 Correct “Hydroxy Appetite”. Consider a title of the sub-chapter “Antimicrobial Activity of Chitosan and Hydroxyapatite Nanoparticles”

16) Line 367 “while MBC of CSNPs on S. sangui was 8 μg/ml”, typo, should be sanguis

17) Line 373 The caption to Figure 11 needs corrections

18) Line 386 a typo in Figure 12 “MIC (µ/ml) Hydroxyapatite nanoparticles” should be µg /ml

19) The Figure should be re-drawn/reformatted. Design of Figures 11 and 12 should be unified.

20) Line 391 typo, should be “microscopy”

21) Line 419 The caption needs to be re-written

22) Line 423 Discussion – in need of language improvements to be done by a native English speaker

23) Line 485 Consider using “concluded” instead of “have investigated”

24) Line 500 typo “implants” instead of implant

25) Reference 20 is missing (line 569)

Reviewer 2 Report

Authors have addressed concerns.
